# Lightweight ML-based Runtime Prefetcher Selection on Many-core Platforms

Erika S. Alcorta*‡, Neeraja J. Yadwadkar*†, Andreas Gerstlauer*

*The University of Texas at Austin. *esalcort@utexas.edu,neeraja@austin.utexas.edu,gerstl@utexas.edu*

‡Ampere Computing.

†VMWare Research.

*Abstract*—**Modern computer designs support composite prefetching, where multiple individual prefetcher components are used to target different memory access patterns. However, multiple prefetchers competing for resources can drastically hurt performance, especially in many-core systems where cache and other resources are shared and very limited. Prior work has proposed mitigating this issue by selectively enabling and disabling prefetcher components during runtime. Traditional approaches proposed heuristics that are hard to scale with increasing core and prefetcher component counts. More recently, deep reinforcement learning was proposed. However, it is too expensive to deploy in real-world many-core systems. In this work, we propose a new phase-based methodology for training a lightweight supervised learning model to manage composite prefetchers at runtime. Our approach improves the performance of a state-of-the-art many-core system by up to 25% and by 2.7% on average over its default prefetcher configuration.**

## I. INTRODUCTION

Hardware data prefetching can reduce memory latency and significantly improve the performance of many applications, provided it accurately and promptly detects their memory access patterns. However, individual prefetchers typically target specific or limited sets of patterns [12]. To address this limitation, modern processors combine multiple prefetcher components, thus covering a wider range of access patterns than monolithic prefetchers [12]. Increasing the number of prefetches in the system can lead to higher contention and pollution of shared resources like memory bandwidth and cache space [9], [14]. Furthermore, in multi-core systems, enabling prefetching can sometimes hurt performance depending on the workload [10]. Consequently, modern processors offer users the ability to adjust prefetcher components through registers [7], [14], but selecting when to enable or disable prefetcher components for any program application is a challenging task.

The variety of dynamic workload behaviors in program applications is very large, and the best prefetcher selection may change depending on the workload behavior. For example, Fig. 1 shows the execution time of 10 programs from the SPEC CPU Int Rate 2017 multi-programmed benchmark suite [18] running on a many-core hardware platform with three different prefetcher configurations: *ON*, *OFF*, and *Def*. *ON* enables all prefetcher components; *OFF* disables all prefetcher components; and, *Def* sets the default configuration, which enables one prefetcher. The figure depicts that the best selection is different for each program. While this example only compares three configurations, modern systems offer

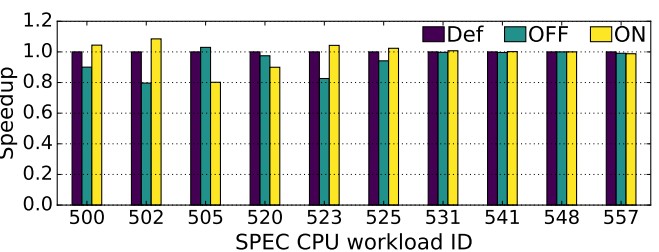

Fig. 1. Speedup (higher is better) of SPEC CPU Int Rate 2017 benchmarks with all prefetchers disabled (OFF) and all prefetchers enabled (ON) compared to the default config (Def). The best configuration depends on the workload.

more options, increasing the complexity of runtime decisions that map workload behaviors to prefetcher selection.

Previous research has explored various techniques for tuning prefetcher components at runtime to maximize performance, a task commonly known as runtime adaptive prefetching. Some studies use heuristics and explore all or a subset of configurations during program execution to make decisions [11], [14], [15]. However, exploring configurations during runtime misses performance opportunities and does not scale with increasing configurations and core counts. More recent works have used machine learning (ML) models to train a policy offline and evaluate it online [7], [9]. However, they do not provide sufficient proof that their models are generalized enough to handle unseen workloads, and their proposed models are too expensive to implement on a real-world platform. Furthermore, none of these runtime adaptive prefetching studies have investigated many-core platforms, which present unique challenges that do not manifest at lower core counts.

In this work, we propose a runtime prefetcher selection approach that uses a low-overhead machine learning model to enable or disable prefetcher components based on their expected performance improvement on a state-of-the-art many-core platform. We collect hardware counter data to monitor the system workload and propose a new methodology that uses phase classification [1] and supervised learning to correlate workload phases with the best selection of prefetcher components. We demonstrate the effectiveness of our approach by deploying a software-based version on a state-of-the-art cloud-scale hardware platform. Our approach can also be implemented in hardware on future processor designs.

We summarize the contributions of this paper as follows:

1) We propose phase classification to group similar workload behaviors and find the best prefetcher selection for each phase using a supervised learning formulation.
2) We implement a decision tree model that is lightweight, requiring only 42 bytes of storage, yet accurate enough to improve the execution time of cloud workloads running on a 160-core AmpereOne, a state-of-the-art many-core platform.
3) We demonstrate our model's ability to generalize and improve the performance of workloads that were not seen during training. Our evaluation includes data collected from diverse multi-programmed and multi-threaded workloads. Our results show that our model can improve the performance of new workloads by up to 25% over the platform's default prefetcher and by 2.7% on average.

## II. RELATED WORK

Prior work has proposed numerous approaches to reduce the contention generated by prefetchers in multi-core systems. Some work is concerned with extending the design of prefetchers [3]–[6], [13], [16], [19] while others have proposed prefetcher-aware cache insertion and eviction policies to manage cache contention [8], [19]. While these solutions focus on tuning an individual prefetcher, our approach is concerned with managing the components of composite prefetchers.

Various studies in composite prefetching management propose heuristics to select prefetchers at runtime [11], [14], [15]. These approaches study different metrics to rank prefetcher configurations based on performance [11], [15] or other heuristics [14]. The ranking is obtained during execution time by performing an exhaustive search that executes every prefetcher configuration for one sample. The best-ranked configuration is selected for a pre-determined period of time. This process is repeated after either a fixed time window [14] or a phase change, defined by a fixed percentage change in system performance [11] or annotated in code [15]. However, exhaustively searching multiple configurations during runtime is not scalable as the number of prefetchers and applications increases. Additionally, the time spent searching necessarily misses optimization opportunities. Lastly, ranking prefetcher configurations based on the performance of a single sample fails to acknowledge short-term performance variations in workloads [1], which may lead to selecting the wrong configuration.

More recent work has introduced ML-based composite prefetcher management approaches. These models eliminate the need to search the configuration space exhaustively by learning to generalize from fewer samples. In [7], the authors proposed formulating the problem with contextual bandits. They train one model per prefetcher component while other prefetchers are always on. However, they do not evaluate the coordination of prefetchers, since the models are not enabled simultaneously. Additionally, they found that they never need to disable some prefetchers in their quad-core system. This is not the case in many-core systems, where it is

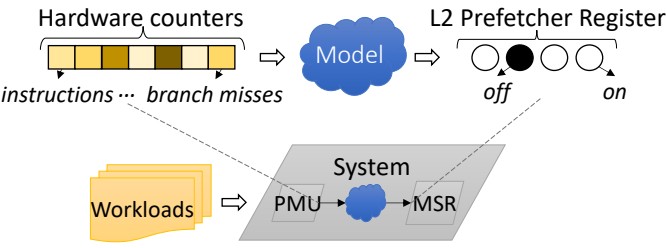

Fig. 2. Prefetcher selection overview.

sometimes beneficial to disable all prefetchers, as was shown in Fig. 1. In [9], the authors propose using deep reinforcement learning (RL) to coordinate multiple prefetchers. However, deploying deep RL models on real-world systems is very expensive in terms of training, power, storage, and latency costs. In contrast, we propose a supervised learning model with minimal costs that can be either implemented in existing runtime management systems or easily deployed in hardware. Moreover, these studies [7], [9] only considered multi-programmed workloads and did not investigate whether their models can improve the performance of unseen (i.e., not used for training) program applications. Our work demonstrates that our proposed lightweight runtime prefetcher selection model can generalize its predictions to unseen and multi-threaded workloads.

## III. PREFETCHER SELECTION MODEL DESIGN

The task of selecting a prefetcher configuration during runtime with a model is represented in Fig. 2. The model aims to map a vector of hardware counter values into a prefetcher selection decision. We collect hardware counters by accessing the performance monitoring units (PMU) of the system and set the prefetcher decision through a model-specific register (MSR). This section outlines our proposed method for designing and training such a model. We start by introducing the problem formulation, followed by an explanation of our approach, which involves both offline analysis and online implementation.

### A. Problem Formulation

The goal of a prefetcher selection policy is to minimize the execution time of a workload, which we define as $G$. The execution of a workload is represented by a trace of hardware counters, $U \in \mathbb{R}^{T \times C}$, where $T$ is the number of samples and $C$ is the number of collected hardware counters. An observation of $U$ at time $t$ is represented as $U_t$. The hardware counters are transformed into features $X_t = \Omega(U_t), X_t \in \mathbb{R}^M$, where $M$ is the number of features. For example, this transformation $\Omega$ includes calculating the IPC with the *instructions* and *cpu-cycles* hardware counters. We use $\rho_t$ to represent the IPC of a sample at time $t$, $\rho_t \in X_t$. We partition the goal of minimizing the execution time into smaller goals that maximize the IPC of each sample, $\rho_t$, based on the observation that the average IPC is inversely proportional to the execution time.

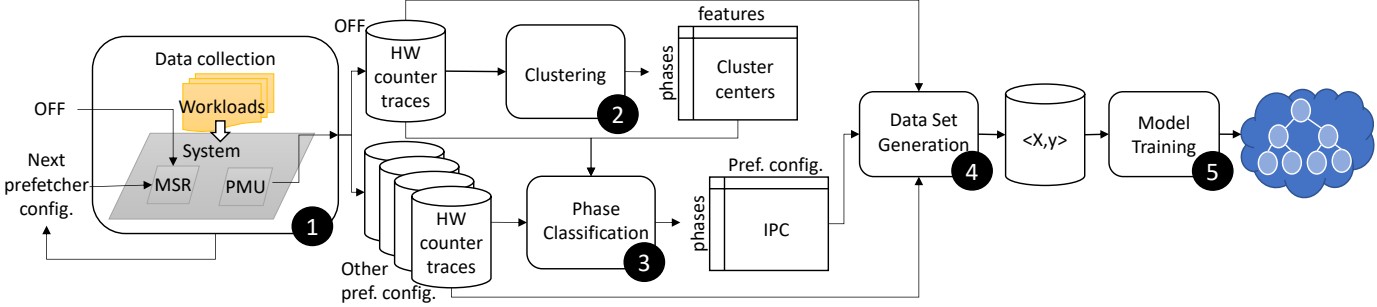

Fig. 3. Proposed analysis to generate our runtime prefetcher selection model.

TABLE I
LISTS OF COLLECTED HARDWARE COUNTERS AND FEATURES.

| Hardware counters ($U$) | Features ($X = \Omega(U)$) |
|---|---|
| Instructions | Instructions per cycle (IPC) |
| Memory accesses | Memory accesses per kilo instructions |
| Branch misses | Branch misses per kilo instructions |
| Cache misses | Cache misses per kilo instructions |
| CPU cycles | Cache misses to memory accesses ratio |
| L2 data cache refills | L2 data cache refills to cache miss ratio |
| L2 instruction cache refills | L2 instruction cache refills to branch misses ratio |

At each time step $t$, a machine learning model, $f$, predicts an output, $y_{t+1}$ based on the features $X_t$ with the goal of maximizing $\rho_{t+1}$. The output is a one-hot encoded vector, $y_t \in \{0, 1\}^N$, where $N$ is the number of prefetchers, and each element in the vector indicates whether the prefetcher should be enabled or disabled.

*B. Data Analysis and Model Training*

After partitioning our goal of minimizing a workload's execution time into smaller goals that maximize the IPC of each sample of the workload, we need to define a ground truth in order to train a supervised learning model. We propose a method that analyzes data and generates labels to train a runtime prefetcher selection model in an offline fashion. Our method is depicted in Fig. 3, comprising five stages detailed below.

*1) Data Collection:* We periodically collected hardware counter data from different workloads to later train our model. For each workload, we collected one trace of hardware counters per prefetcher configuration.

*2) Clustering:* In order to compare the samples of different prefetcher configurations, we propose clustering similar PMU behaviors together to find phases within the workloads. Our methodology involves training a clustering model with data from only one prefetcher configuration. We chose *OFF* as our baseline since it shows workload behaviors without the effects of prefetching. We scaled all features to a range between 0 and 1 using a min-max scaler and clustered all the workload traces of the baseline configuration using *k-means*. This produces a table of cluster centers, which is then used for phase classification.

*3) Phase Classification:* Once the cluster centers have been generated using data from the baseline configuration, we use

them to classify the phases of data samples in all traces. Next, we group all samples in the same phase and prefetcher configuration and calculate the average IPC per phase. This allows us to compare the performance of different prefetcher configurations across workload phases.

*4) Training Set Generation:* We use the phase classification labels to determine the best prefetcher configuration for each sample, which we define as the configuration that yields the highest average IPC for the corresponding phase. We consider this definition as our ground truth. Associating each sample and its phase classification with the best prefetcher configuration generates a supervised training set that assigns each sample's features $X_t$ to the ground truth prefetcher selection, $y_t$.

*5) Model Training:* We use our generated data set to train a decision tree model. We found that it only needs four input features instead of seven while maintaining high prediction accuracy. This reduces the number of hardware counters that we need to collect during runtime.

*C. Runtime Implementation*

We implemented our prefetcher selection model as a program with a thread that is invoked every 100 ms. The thread accesses hardware counter values using *perf's* system call. Then, it transforms the counters into features and performs inference on the decision tree. Finally, it writes the decision tree output to the corresponding fields in the prefetcher MSR.

## IV. EXPERIMENTAL RESULTS

We collected data from one multi-programmed benchmark suite, SPEC CPU Int Rate 2017 [18], and two multi-threaded Java benchmark suites, DaCapo [2] and Renaissance [17], to evaluate our approach. We use SPEC CPU workloads for training and validation and DaCapo and Renaissance for testing. All workloads run on AmpereOne, a cloud-scale many-core platform with 160 ARMv8.6+ ISA cores, 2MB of L2 cache per core, 64MB of system-level cache, and 256GB of DDR5-4800 memory running Fedora Linux 36. The platform has 12 different prefetcher configurations, which can be tuned with a hardware register. For each prefetcher configuration, we collected one trace of hardware counters per workload, resulting in a total of 120 traces (12 prefetcher options × 10 workloads). Each trace consisted of $C = 7$ hardware counters

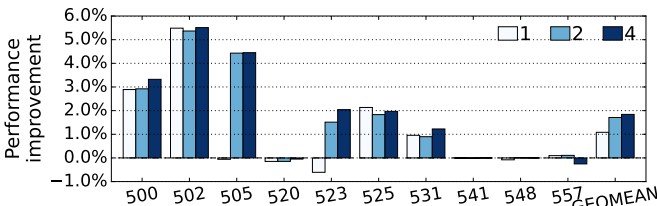

Fig. 4. Performance improvement (execution time reduction) of different decision tree model depths over the default prefetcher on SPEC CPU benchmarks.

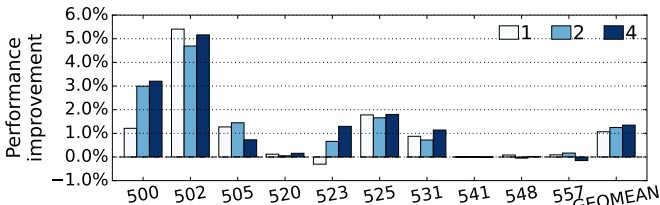

Fig. 5. Performance improvement (execution time reduction) of different decision tree model depths over the default prefetcher on SPEC CPU benchmarks compiled with gcc-12.

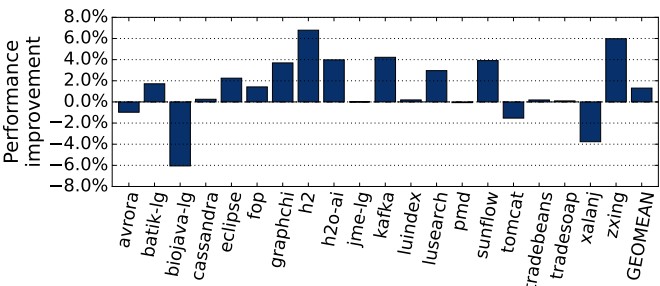

Fig. 6. Performance improvement (execution time reduction) of a runtime prefetcher selection tree of depth 4 trained on SPEC CPU over the default prefetcher on DaCapo benchmarks.

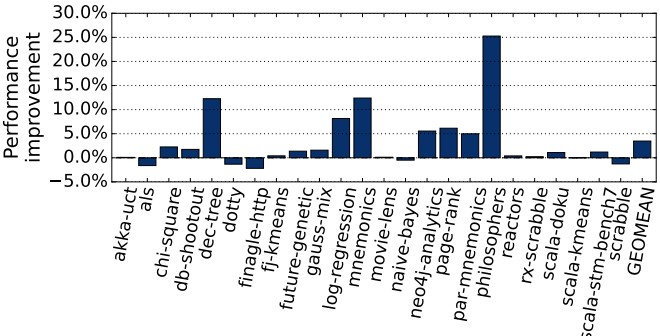

Fig. 7. Performance improvement (execution time reduction) of a runtime prefetcher selection tree of depth 4 trained on SPEC CPU over the default prefetcher on Renaissance benchmarks.

collected periodically every 100 ms with Linux's *perf* tool. The hardware counters were transformed into $M = 7$ features. See Table I for the lists of hardware counters and features.

Fig. 4 shows the speedup of all SPEC CPU benchmarks when prefetcher selection is enabled and exploring the decision tree depth hyperparameter with depths of 1, 2, and 4. The results are normalized to the system's default prefetcher. The geomean is shown on the right side of the plot. On average, enabling system-wide runtime adaptive prefetching improves the performance of SPEC workloads by 1.9% and up to 5.5% in the best scenario.

We want to measure the ability of the model to improve performance even with system changes. For this test, we evaluated our models on the same programs but with different binaries. Specifically, we recompiled SPEC CPU benchmarks with a different compiler, gcc-12, which introduces several new code optimizations when compared to previous versions, such as improved vectorization and structure splitting. So although the same work is completed, the data access patterns may vary widely as in the case of 505.mcf. Then, we enabled prefetcher selection with the same models that were previously trained on SPEC programs compiled with gcc-10. The results are shown in Fig. 5. The best-performing decision tree has a depth of 4. We observe a similar performance improvement trend between the gcc-10 and gcc-12 experiments and demonstrate that the model still improves performance even when presented with different binary files.

We further test the performance of our model by presenting it with completely new workloads (not used for training). We ran workloads from the DaCapo and Renaissance suites. Note that in addition to being new workloads, they are multi-threaded instead of multi-programmed, written in a different language (Java), and compared to SPEC CPU they spend more time in operating system code, network stack, and

synchronization (locking and snooping). We tested our best-performing decision tree with depth 4 on each suite and show our results in Fig. 6 and Fig. 7. For most of the workloads, dynamic prefetcher selection reduces the execution time, with the best scenario being 25%. However, as opposed to SPEC CPU results, some programs lose performance. Nonetheless, the geomean performance improvements for DaCapo and Renaissance suites are 1.7% and 3%, respectively. The improved performance of all these unseen workloads together is 2.7%.

A major benefit of our proposed model, as opposed to prior work, is the lightweight implementation. The decision tree has a maximum depth of 4. It requires storing 15 nodes with two parameters each: the feature ID (in our case, 2 bits for four features) and the compare value (we use 16 bits but can be reduced to 8 bits). Additionally, the eight leaf nodes require storing the prefetcher selection when true or false (4 bits in our case). The total size of our model is only 42 bytes, which makes it easy to fit on any embedded firmware or hardware deployment.

## V. CONCLUSION AND FUTURE WORK

We proposed a lightweight model for runtime prefetcher selection for many-core platforms. It can improve the performance of unseen workloads by up to 25% and 2.7% on average over the default prefetcher.

These early results suggest that runtime prefetcher selection can be formulated as a workload-agnostic offline supervised learning problem; however, further investigation is required to

determine why it performed poorly in a few benchmarks. The investigation should determine whether the problem is training coverage, i.e., the input features are in a different distribution from the training set, or the problem is workload specific, i.e., for the same set of input features, the best prefetcher selection is different depending on the running program. Our proposed approach estimates the best prefetcher selection for all the cores in the system. Future work includes investigating lightweight runtime prefetcher selection that is more practical for per-core decisions.

## ACKNOWLEDGEMENTS

We thank Mahesh Madhav and Scott Tetrick who played a vital role in the success of this research project. This work was supported in part by Ampere Computing and NSF grant CCF-1763848.

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
