# OpenReview forum: "Lightweight ML-based Runtime Prefetcher Selection on Many-core Platforms"
_iscaconf.org/ISCA/2023/Workshop/ASSYST — ASSYST Oral_

### Official Review · Reviewer_83Yz · 2023-05-02
**Lightweight decision tree model to dynamically select HW prefetchers. Well-motivated and executed paper with some gaps to fill.**

**Rating:** 6
**Confidence:** 4

**Review:**

## Summary of paper:
This paper presents an end-to-end workflow to generate and deploy a supervised ML (decision tree) model to dynamically select between prefetched configurations in a many-core (NoC) system. Recent work has shown the benefits of using composite prefetchers, consisting of multiple algorithms/implementations specialized for a specific access pattern, compared to a monolithic prefetcher that cannot easily generalize. This motivates the need for a controller to select which prefetcher is active at a given time — a non-trivial problem. The authors motivate the impact of a well-tuned controller, showing that the optimal prefetcher varies across workloads. The authors formulate this problem as an optimization problem to minimize workload execution time (i.e., maximize IPC), where the inputs are traces of hardware counters (from CSRs) and the output is a configuration for each prefetcher (on/off) at the next time step. The authors propose an offline training pipeline, which generates a training set based on classifying the phase of all samples across traces (and ground truths based on observed IPC). They train a lightweight decision tree and deploy it to tune a NoC every 100ms. The authors show 1.9% improvement on average on SPEC, and also evaluate using a new binary and unobserved workloads.

*Explicit strengths and weaknesses listed in next section.*

## Detailed Comments
I think the problem that this paper tackles is certainly important. I really like the introduction/motivation (Figure 1) that sets the importance and potential impact of such a problem. I believe that the overall approach of training a light-weight model that can be deployed into real hardware is a promising research direction.

There are various suggestions which I think could significantly strengthen future versions of the paper (and would be critical for a full-length paper), but given the venue and limited pages the authors have, these are not fatal.

- My main question is in the “offline analysis” component of the work. From the introduction/motivation, I assumed that this paper was going to head towards proposing an online training mechanism (especially considering III.B is titled “Data Analysis and Online Training”), where the model would be continuously trained as it is deployed. This certainly makes sense, as the authors motivated the need for a lightweight model (e.g., something that could be trained and deployed using libraries such as Vowpal Wabbit) and a model that could react to unseen workloads. However, Fig 3 and the text seem to imply that an offline dataset is created, the model is trained offline, and then the model is deployed. The only “online” component I see is inference, which doesn’t seem that unique. This is similar in fashion to works that the authors have cited [6, 8]; is the small model size the main difference? It would be really interesting to see if the training pipeline could actually be deployed in online training, and if not, why? You give yourself 100ms per action; retraining a lightweight classifier sees doable in that time. The text needs to be more clear about if and how online training is used.

- Some more evaluation results would certainly be helpful. I see two main directions.
    - First, your motivation showed significant gains with a pretty simple configuration (off/default/on). For example, SPEC 502 showed a 25% gain. However, in the evaluation, the wins don’t live up to the hype; 502 achieved only 5%. Do you have an understanding of why this is? It would help if you compared to some other baselines (e.g., an oracle, or a more heavy-weight model), and analyzed why your model made different choices. Is 2.7% enough to call this a success?
    - Secondly, I don’t fully understand the justification behind figure 5. I’m not fully convinced that moving from gcc 10 to gcc 12 will change access patterns that much to need an adaptable prefetcher. Please either justify this, or potentially explore other options (e.g., would changing optimization flags yield a more impactful result?).

- I’m not entirely certain how NoCs come into play here, especially considering the focus on NoC platforms. Is it as simple as more cores = more combinations of prefetchers? Are there any unique challenges in NoCs?

- The proposed future work is promising, and would address some of my questions.

- There’s a line of work from Google on applying ML to prefetching (Learning Memory Access Patterns @ICML’18 and followup work). It would be good to discuss that in context to your work.

- Minor nits:
    - Define what a phase is earlier
    - Is figure 1 normalized to default? It’s unclear from the text




**Review (Strengths/Weaknesses):**

## Strengths:
- A well motivated problem. There is a need for an optimized, light-weight control algorithm for prefetcher selection especially in modern systems which use composite prefetchers.
- Sound training pipeline. The problem statement is well-formulated, and the authors demonstrate a concrete understanding of the steps and constraints needed to deploy such a system (collecting data via hardware counters, configuring PMUs, need for a small model able to deploy into real hardware).
- Demonstrates that the model is effective on a wide set of benchmarks, including unseen workloads.

## Weaknesses
- Unclear description of training methodology, e.g., can this be applied to online training?
- Provides somewhat incremental improvements; more analysis would better help justify the results
- Lack of exploration and comparison to other methods (e.g., more expensive offline models)


**Reviewer Expertise:**

Knowledgeable: I used to work in this area and/or I try to keep up with the literature but might not know the latest developments.

---

### Official Review · Reviewer_HwvZ · 2023-05-06
**Lightweight ML-based Runtime Prefetcher Selection on Many-core Platforms**

**Rating:** 6
**Confidence:** 4

**Review:**

This paper describes the use of a light-weight, decision tree model to select a performant combination of prefetchers to enable at runtime. The authors demonstrated the importance of prefetcher selection using SPEC benchmarks and illustrated a supervised learning methodology to train the decision tree model. The outcome is an average performance improvement of 2.7% over the default prefetcher running SPEC.

**Review (Strengths/Weaknesses):**

Strengths:
* The author shows the robustness and the generality of their approach by running the decision tree against new and different varieties of workloads that have different characteristics (compiled differently, multi-programmed vs threaded).
* The paper is well written with clear problem statements, description of the approach and results. The author addresses how they resolved the key challenges of phase classification and data generation.
* The implementation is very lightweight (in terms of memory and computation cost), making it amenable to hardware integration.

Weaknesses
* Comparisons are only against the default prefetcher. Some comparisons against heuristics and RL based approaches would make the paper stronger.
* Given there are a limited number of prefetcher configurations (12 as mentioned), it would be interesting to do an exhaustive sweep of these configurations to determine the maximum achievable speedup and compare the decision tree approach against the performance ceiling. It is difficult to judge whether the achieved 2.7% speed up is considered “good”, “amazing” or “just average” in the context of prefetcher selection.
* The author claims their method scales well with increased configuration space and core count, but there is no result to back up this claim.
* Insights into _why_ and _how_ the speedups were achieved by the selection of prefetchers would add to the understandability and explainability of the approach.


**Reviewer Expertise:**

Knowledgeable: I used to work in this area and/or I try to keep up with the literature but might not know the latest developments.

---

### Official Review · Reviewer_pLUi · 2023-05-07

**Rating:** 5
**Confidence:** 4

**Review:**

# Summary

This paper proposes a supervised ML model for enabling and disabling prefetcher components during runtime.


# Comments

1. “We demonstrate the effectiveness of our approach by deploying a software-based version on a state-of-the-art cloud-scale hardware platform.”
* It is hard to evaluate the efficiency of the proposed method as more information is required regarding the default prefetcher and the cloud platform.
2. Figure 1 -> ‘Fig. 1’: recommended to use \cref{Label_A} instead of Figure~\ref{} in latex.
3. Has the decision tree model been selected as the classification method? How did the authors configure the hyper-parameters of the decision tree classifier? I highly recommend authors use an automatic technique, such as Auto-sklearn, to select and fine-tune the parameters of the classification method.
* Feurer, Matthias, et al. "Auto-sklearn 2.0: The next generation." arXiv preprint arXiv:2007.04074 24 (2020).
4. As always, to guarantee the reproducibility of results, I highly recommend authors release the code of this paper.


**Review (Strengths/Weaknesses):**

##	Strengths
*	It is a relevant topic for the conference.
*	The message of the paper is well-supported.
*	Overall, the paper is well-written and easy to follow.

##	Weaknesses
*	Improvements are marginal (2.7% on average over its default prefetcher).
*	Limited experiments: (1) Reporting the accuracy of the proposed classifier is needed; (2) There is no comparison with related studies.


**Reviewer Expertise:**

Knowledgeable: I used to work in this area and/or I try to keep up with the literature but might not know the latest developments.